# An Empirical Study of the Impact of Systems Thinking and Simulation on Sustainability Education

**Caroline Green \*** , **Owen Molloy and Jim Duggan**

School of Computer Science, National University of Ireland Galway, H91 TK33 Galway, Ireland;
owen.molloy@nuigalway.ie (O.M.); james.duggan@nuigalway.ie (J.D.)
\* Correspondence: caroline.green@nuigalway.ie

**Abstract:** Education for sustainable development (ESD) is considered vital to the success of the United Nations' sustainable development goals. Systems thinking has been identified as a core competency that must be included in ESD. However, systems thinking-orientated ESD learning tools, established methods of the assessment of sustainability skills, and formal trials to demonstrate the effectiveness of such learning tools are all lacking. This research presents a randomised controlled trial ($n = 106$) to investigate whether an innovative online sustainability learning tool that incorporates two factors, systems thinking and system dynamics simulation, increases the understanding of a specific sustainability problem. A further aim was to investigate whether these factors also support the transfer of knowledge to a second problem with a similar systemic structure. The effects of the two factors were tested separately and in combination using a two-by-two factorial study design. ANOVA and related inferential statistical techniques were used to analyse the effect of the factors on sustainability understanding. Cohen's d effect sizes were also calculated. Simulation alone was found to increase ESD learning outcomes significantly, and also to support the transfer of skills, although less significantly. Qualitative feedback was also gathered from participants, most of whom reported finding systems thinking and simulation very helpful.

**Keywords:** education for sustainable development; ESD; systems thinking; system dynamics; simulation; transfer of skills; effectiveness; randomised controlled trial; RCT; factorial study; ANOVA

## 1. Introduction

Sustainability has become an increasingly important topic over the last several decades as the harmful long-term consequences of unsustainable human activities and lifestyles have become ever more apparent [1,2]. According to Sverdrup, 'The global sustainability challenges of the future...can only be addressed and solved through a full systems approach' [3].

Education for sustainable development (ESD) aims to equip learners with the skills necessary to reason about complex sustainability problems and to take action to help create sustainable solutions. ESD is seen as a 'key enabler' for the success of the UN's sustainable development goals (SDGs) [4].

Systems thinking has been identified as a core competency in sustainability understanding [5], but it is challenging to teach and learn. Innovative system-orientated ESD learning tools, evaluated for their effectiveness, are needed [6]. Not only are such tools currently lacking, but there is also a lack of widely accepted and validated assessment instruments to evaluate the effects of ESD initiatives [7]. There is also a need for formal trials to evaluate the effectiveness of different ESD approaches [8].

The randomised controlled study described in this article seeks to address these gaps by evaluating the effect of systems thinking on ESD learning outcomes, using a novel learning tool and an experimental framework for assessment.

The literature review section first provides an overview of the concepts of systems thinking, sustainability and sustainability education, and discusses the need for systems thinking and the significant contribution that the system dynamics field has made and can make to sustainability and environmental and systems education.

The next section, entitled 'The Sustainability Learning Tool: Design and Implementation', describes the systems-orientated ESD learning tool built to provide the learning intervention evaluated in the study. Important background research themes are outlined, including the identification of key tools to teach systems thinking, key competencies necessary for sustainability understanding, and current best practice for simulation-based learning environments (SBLEs). The chosen case studies are then described and analysed, and the sustainability learning points derived to form the basis for the assessment of learning outcomes used in the study.

The subsequent sections describe the empirical study in detail: the methodology employed, the results, and, finally, a discussion and conclusions.

## 2. Literature Review

### 2.1. Systems and Systems Thinking

A system is a complex collection of parts interacting to create often counter-intuitive dynamic behaviour [9]. Humans struggle to reason about complex systems [10,11]. Systems thinking is a skill set and a way of thinking that equips people to understand dynamic complexity [12].

### 2.2. Sustainability and Sustainability Education

The concept of sustainability is complex and is often used in an imprecise [13] or even misleading [14] way. Tracing the origins of the term, since the 1980s the UN has been instrumental in developing the related concepts of sustainability, sustainable development and sustainability education. It founded the World Commission on Environment and Development (WCED) in 1980 which was responsible for the influential 1987 Brundtland Report [15]. The definition of sustainability in that report is the one most frequently quoted, namely that 'Sustainable development is development that meets the needs of the present without compromising the ability of future generations to meet their own needs'.

The UN also led efforts to formulate concrete targets for action towards sustainability. In 2000, the UN defined the eight Millennium Development Goals (MDGs) for 2015, of which goal seven was 'To ensure environmental sustainability'. The MDGs were developed further in the 17 Sustainable Development Goals (SDGs), set in 2015 and to be achieved by 2030, and adopted by all 193 United Nations member states.

The UN adopted the Decade of Education for Sustainable Development (DESD) from 2005 to 2014. Education for Sustainable Development (ESD) is explicitly recognized in the SDGs as part of Target 4.7 of the SDG on education. It is seen as 'crucial for the achievement of sustainable development' [5] (p. 63). The Council of the European Union sees ESD as 'essential for the achievement of a sustainable society and is therefore desirable at all levels of formal education and training, as well as in non-formal and informal learning' (Council conclusions on education for sustainable development. https://www.consilium.europa.eu/uedocs/cms_data/docs/pressdata/en/educ/117855.pdf, accessed on 9 June 2021). Thus, ESD is seen as a form of lifelong learning, and necessary for all citizens. It underpins public participation in environmental and developmental decision making [16].

Sustainability education seeks to address the considerable challenge of training learners not only to solve or understand existing complex problems, but also to equip them with skills that they can transfer to new problems as they arise. In the last few years, there has been an urgent call for innovative sustainability pedagogies [17] (p. 58).

O'Flaherty and Liddy provide a useful summary of the approaches so far taken in ESD, including blended learning, drama, simulation exercises, multi-media, problem-based learning and discussion forums [8]. They describe methodological and pedagogical

questions that remain open and highlight the need for assessment frameworks and formal trials for evaluating the effectiveness of different approaches to ESD.

### 2.3. The Need for Systems Thinking in Sustainability Education

Sustainability education is an emerging field. In her review, Maria Hofman-Bergholm explores reasons for problems with its implementation [18]. She finds that systems thinking is required to comprehend the intricate connections in sustainable development [19] (p. 27). Complex reasoning skills must be taught, as they are not inherent. Humans have well-known limitations in cognitive ability to reason about complex systems that must be overcome [20] (p. 599).

Similar observations have been made in the related field of ocean literacy [21]. A pilot study we conducted to investigate the effectiveness of a systems-orientated online ocean literacy learning tool gave promising results [22].

The low level of systems thinking found in teachers contributes to problems with implementing ESD [19]. The SysThema research group (Systems Thinking in Ecological and Multidimensional Areas. https://www.researchgate.net/project/SysThema, accessed on 24 December 2021), with collaborators including Werner Rieß, have investigated how to foster systems thinking in student teachers, since 'science teachers who are required to teach ESD-relevant topics should be proficient in systems thinking and be able to transfer that knowledge effectively to their students' [23]. Orit Ben-Zvi Assaraf has studied the development of systems thinking skills in earth system and biology students [24,25]. Another study reported partial success with attempts to teach systems thinking skills for environmental education [26].

According to Frisk and Larson, sustainability education will only be effective if it incorporates systems thinking, long-term thinking, collaboration and engagement, and action orientation [27]. Sustainability, they say, is fundamentally a call to action, and sustainability education therefore requires experiential, practical and flexible learning methods.

According to Wiek et al., 'Sustainability education should enable students to analyse and solve sustainability problems' [28] (p. 204). This requires a particular set of interlinked and interdependent key competencies. Wiek et al. review the literature and identify five key competencies, the first being systems thinking competence (the others are anticipatory, normative, strategic and interpersonal competence).

According to Soderquist and Overakker, the discipline of systems thinking provides a process, set of thinking skills and 'technologies' that can improve the systemic understanding that is required for sustainability education [29]. These include stock and flow mapping, computer simulation, and simulation-based learning environments. They claim that simulation-based learning environments build mental simulation capacity, if they are designed carefully.

Cavana and Forgie describe a number of well-established systems education programs and review teaching approaches for sustainability education [6]. They explore the strong links between systems approaches and sustainability goals, illustrating that the two are so entwined as to be inseparable. They describe the need for, and the lack of, simulation-based learning environments for systems thinking-orientated sustainability education. A brief review of contributions from the field of system dynamics is useful to address this need and follows next.

### 2.4. Insights and Relevant Work from the Field of Systems Dynamics

A substantial body of knowledge focused on the modelling and simulation of complex human–environmental systems has accumulated in the field of system dynamics since the 1970s. This can inform efforts to develop effective, innovative systems-orientated sustainability education tools.

System dynamics modelling was first used to address sustainability in Jay Forrester's 'World3' model, which formed the basis for the influential book, 'Limits to Growth' [30]. There have been many subsequent examples from environmental models [31,32], world

models [3,33], models for water supply, waste management, air quality, land use [34], fisheries [35,36], climate change [37] and its consequences [38], models of social and economic development [39], reindeer pasture management [40], food security [41], marine protected areas [42], models for optimising strategy for achieving the SDGs [43,44], and many more.

Furthermore, the system dynamics community has identified education as a priority for a long time. The Creative Learning Exchange was founded in 1991 by Jay Forrester 'to encourage the development of systems citizens who use systems thinking and system dynamics to meet the interconnected challenges that face them at personal, community, and global levels' (About The Creative Learning Exchange. http://www.clexchange.org/cle/about.asp, accessed on 9 November 2021). They provide resources representing decades of experience of teaching systems thinking and system dynamics for real-world problem solving to school children [45,46].

System dynamics models and simulations have frequently been employed for the purpose of environmental education [47,48]. There are flight simulators for sustainability [49] and climate change [50], simulation-based learning environments to teach sustainability [51,52], and interactive simulation-based games to explore sustainability [53]. System Dynamics models and simulations have also been used to try to explain why renewable resources are so often over-utilised; this is because of faulty reasoning and systematic misperceptions of the dynamics of complex systems [40]. Simulation has been shown to improve understanding and performance in a natural resource management task [54]. Simulation can serve effectively as the 'problem' in problem-based learning [55], and as an experiential activity it can both increase retention and have a stronger influence on behaviour than declarative learning [27] (p. 11).

There is debate about whether simulation based on stock and flow models is an essential, or an advanced part of systems thinking, or an extension of it [56]. According to Richmond, 'System thinkers use diagramming languages to visually depict the feedback structures of…systems. They then use simulation to play out the associated dynamics' [57]. Because simulation is seen as an essential by some [20] (p. 37), [58] (p. 355), but not by all, systems thinkers, in our study, the effect of adding systems thinking and simulation was evaluated separately and in combination.

System dynamics scholars have also long been interested in the transfer of insights between management situations that share common structural characteristics, going back to Forrester [58] (p. 355). According to Sterman, perhaps counterintuitively given the immensely rich and varied complex systems in the world around us, 'most dynamics are instances of a fairly small number of distinct patterns of behaviour' [20] (p. 108). Senge describes Systems Archetypes as 'nature's templates' [59] (p. 92). They reveal an elegant simplicity underlying complex issues. Mastering them represents putting systems thinking into practice. Indeed, Richmond includes what he calls 'generic thinking' in his list of eight critical systems thinking skills [60]. Once an archetype is identified, 'it will always suggest areas of high- and low-leverage change.' For this reason, Kim views archetypes as diagnostic tools [61–63].

Because of the importance of systems archetypes to many systems thinkers, and because of the potential benefits of their use in sustainability education, their effect on the transfer of sustainability skills is explored in this study. According to Gary and Wood, 'Despite the promise [of transfer between structural analogs], limited empirical research on this topic exists in system dynamics' [64]. The choice of two sustainability problems that share a common systems archetype was made to test the hypothesis that learners can recognise similar patterns in different contexts, and therefore transfer their learning. If successful, this approach would make a strong case for a patterns-based approach to sustainability education, which would build systems and environmental literacy, obviating the need to teach each sustainability challenge in a piecemeal fashion.

*2.5. Aims of the Study*

Summarising the themes identified in the reviewed literature, the following research areas were identified and motivated the work described in this paper:

1.  There is a need for systems-based sustainability learning tools that can be shown to increase the effectiveness of sustainability education.
2.  It would be useful to evaluate the effect of systems thinking (theory, tools and techniques) separately from that of interactive simulation, so that the effect of each factor on learning outcomes, and their combined effect, can be compared.
3.  If systems thinking can facilitate recognition of similar systemic structures in different sustainability problems, this could make a useful contribution to the development of transferable sustainability skills.
4.  Formal trials to evaluate the effectiveness of approaches to ESD, including a systems thinking approach, are needed.

**3. The Sustainability Learning Tool: Design and Implementation**

In order to address the gaps identified in the literature, a systems-orientated sustainability learning tool was designed and developed and its effectiveness was tested in the trial described in this paper. The design decisions and features of the learning tool are described in this section.

The learning tool supports a teaching approach that combines two sustainability topics, each supported by a case study, together with relevant systems thinking principles and simulation exercises designed to build understanding of the problem dynamics. The two topics, deer herd management and sustainable fisheries, are both examples of renewable resource management. Each problem is illustrated with a historic case where over-exploitation of the renewable resource led to overshoot and collapse.

An open access version of the learning tool is available online: https://exchange.iseesystems.com/public/carolineb/sustainability-learning-tool/ (accessed on 29 December 2021). It differs from the original version used for research in that it does not require a login and does not collect login or simulation data. Survey answers, although recorded, will no longer be checked, and may be deleted after 30 days. The pre-survey, which included a consent form, has been removed. Learners are no longer randomly allocated to treatment groups, but instead can choose which version of the tool they wish to see.

The learning tool is not a simulation game or a flight simulator, in that learners are not asked to take the role of an actor in the scenarios. The systems thinking and simulation elements in the learning tool offer the 'big picture' of the systems underlying two sustainability problems and offer insights into their essential structure and dynamics. The learning tool also explores sustainable solutions to the problems. The emphasis is thus on systemic understanding and policy making.

Table 1 summarises aspects of the learning tool, and the following sections provide more information where necessary, following the order of the table.

**Table 1.** Elements of the Learning Tool.

| Design Element | Description |
| --- | --- |
| **General description** | • Online, single session, lasting 50 to 100 min in total (including quizzes)<br>• All groups see standard non-systemic introductions to the two main topics using domain-specific terminology<br>• Additional systems thinking and simulation sections are seen by treatment groups |
| **Case Studies** | • Deer Herd management (illustrated with the story of Kaibab deer)<br>• Sustainable fisheries (illustrated with the story of Grand Banks cod fishery collapse) |

**Table 1.** *Cont.*

| Design Element | Description |
| --- | --- |
| **Systems Archetype** | <ul><li>Limits to Growth (Overshoot and Collapse).</li><li>*Applies to both case studies*</li></ul> |
| **System Dynamics model** | <ul><li>The model underlying the simulation exercises was a deer herd population model adapted from Breierova [65] and created in Stella Architect</li></ul> |
| **SBLE design principles** | <ul><li>Case-based learning</li><li>Tasks, exercises and feedback guide use of simulations</li><li>Encourage reflection and promote cognitive dissonance (reconstruct beliefs)</li><li>Model/simulations increase in complexity</li><li>A small model can reveal essential dynamics</li><li>Learner can change some model variables and see graphical results instantly</li></ul> |
| **Sustainability Principles or topics** | Sustainability skills/knowledge:<ul><li>General definition of sustainability</li><li>Sustainable use of renewable resources: limit growth, respect carrying capacity and monitor the system</li><li>Herman Daly principle: Renewable resources must be used no faster than the rate at which they regenerate</li><li>Perform growth calculations and interpret graphs</li><li>Define sustainability in context</li><li>Understand limits and capacity (includes carrying capacity, maximum sustainable herd size, maximum sustainable yield in fisheries, overgrazing and overfishing)</li><li>Dynamic reasoning (including stock and flow reasoning)</li><li>Analyse a complex system (structural causes of dynamic behaviour)</li><li>Make judgments about sustainability (whether a system is sustainable or not)</li><li>Strategies for sustainability</li></ul> |
| **Systems Thinking Principles and tools** | Systems thinking skills/knowledge:<ul><li>Define systems and systems thinking</li><li>Feedback loops</li><li>Causal Loop diagrams</li><li>Behaviour over Time graphs</li><li>Structure determines behaviour</li><li>Stock and Flow diagrams</li><li>Identify common system patterns (archetypes)</li><li>Identify leverage points (places to intervene in a system)</li><li>Understand system equilibrium (a dynamic and sustainable state)</li></ul> |
| **Simulation Exercises** | <ul><li>Simulate deer herd growth in first four years (exponential increase)</li><li>Simulate deer herd growth in first ten years (exponential increase and then decline)</li><li>Simulate deer herd growth, this time with vegetation added to the graph (vegetation decline explains decline in deer population)</li><li>Simulate to find the estimated vegetation level after one year, given vegetation growth and simultaneous consumption by deer (interacting stock levels are hard to calculate without simulation)</li><li>Try lowering initial deer population to avert collapse (this only delays it)</li><li>Try increasing initial vegetation level to avert collapse (this only delays it)</li><li>Try changing deer birth and death rates to obtain a stable population (birth and death rates must be equal)</li><li>Try to make the deer herd sustainable (stabilise deer population AND ensure it does not exceed the carrying capacity)</li></ul> |
| **Platform** | <ul><li>Stella Architect interface published on ISEE Exchange</li><li>Embedded SurveyMonkey quizzes and surveys</li></ul> |

### 3.1. General Description of the Learning Tool

The learning tool consists of two main sections, one for deer herd management and one for fisheries, as shown in Figure 1. The deer section contains additional sections for systems thinking and simulation for some treatment groups.

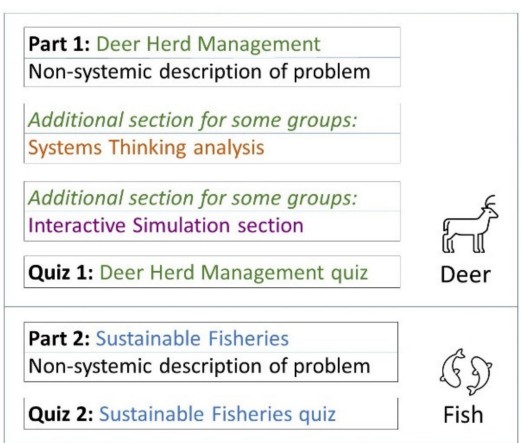

**Figure 1.** Sections of the ESD learning tool.

The standard non-systemic introductory sections in parts 1 and 2 are seen by all groups. Pages consist of text, images, graphs and short embedded video clips. The general concept of sustainability is first explained and explored, then each sustainability theme is described using standard domain-specific terminology. See Figures 2 and 3 for sample pages, one for each case study. These sections each take about 15 min to work through.

The systems thinking (ST) section first explains core ST principles and tools in general terms, and then uses these to analyse the deer herd population dynamics. A sample page from this section is shown in Figure 4. The section is seen by some groups and takes about 30 min to work through.

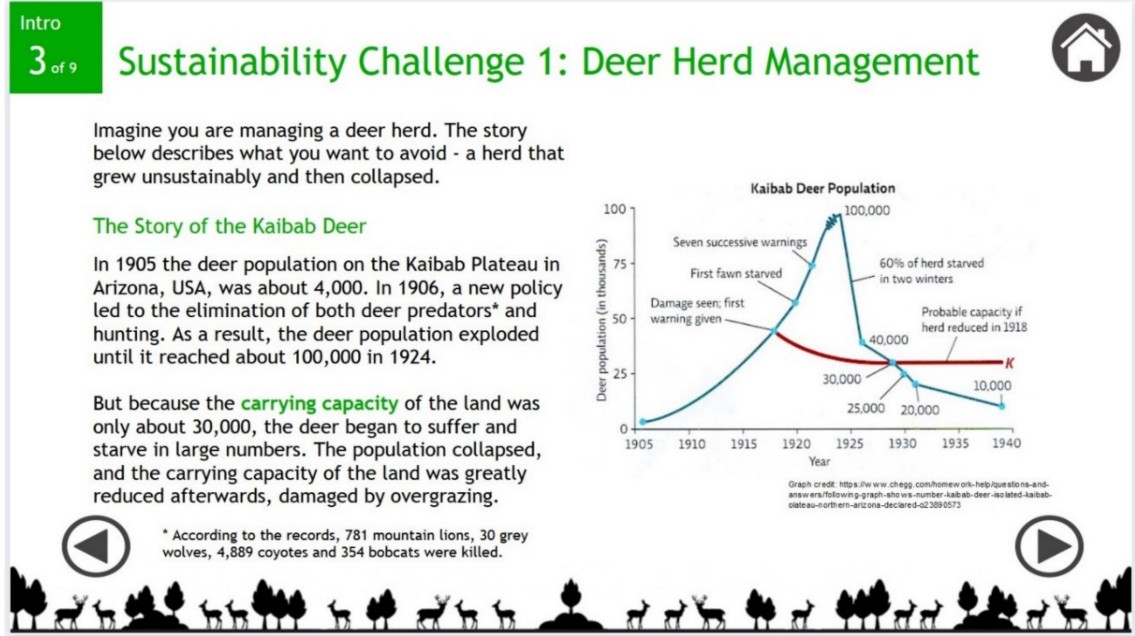

**Figure 2.** Sample screenshot from the introductory section (non-systemic description of the deer herd management problem).

The simulation section is presented using text and embedded simulations, with a pop-up comment, hint or explanation available for each exercise to provide feedback to the learner. A sample simulation exercise is shown in Figure 5. This section is seen by some groups and takes about 20 min to work through.

The learning session, including completion of quizzes and surveys, lasted between approximately 50 min for the control group and 100 min for the full treatment (ST + Sim) group.

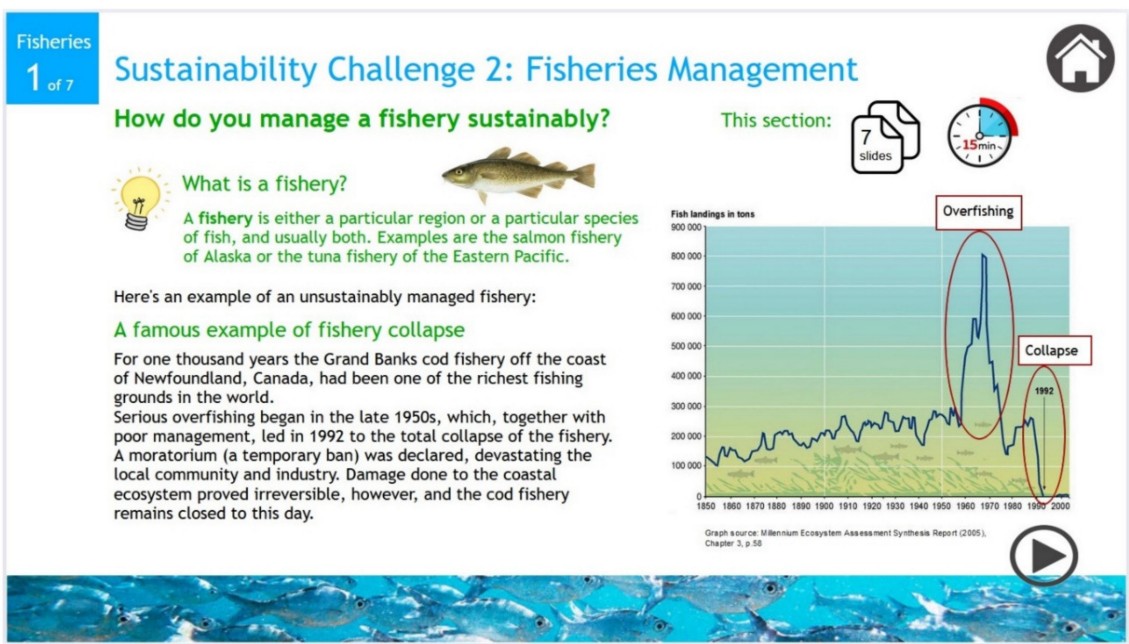

**Figure 3.** Sample screenshot of Part 2 (non-systemic description of sustainable fisheries management).

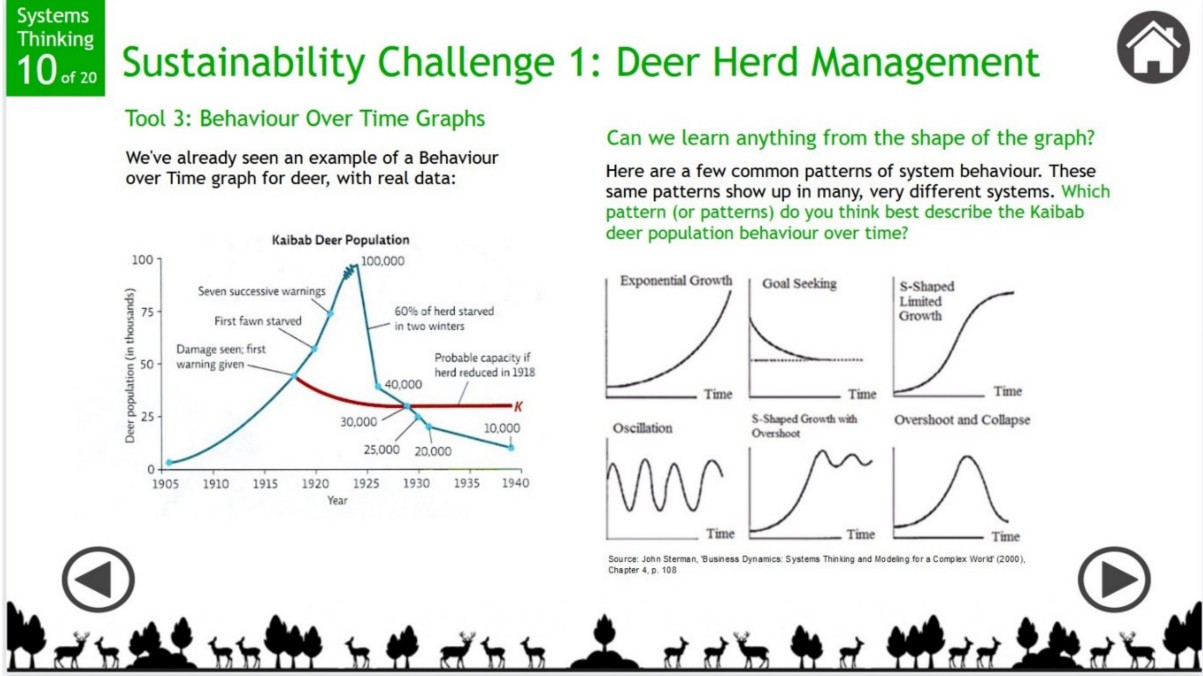

**Figure 4.** Screenshot of a page from the systems thinking section (first page of analysis of behaviour over time graph for Kaibab deer).

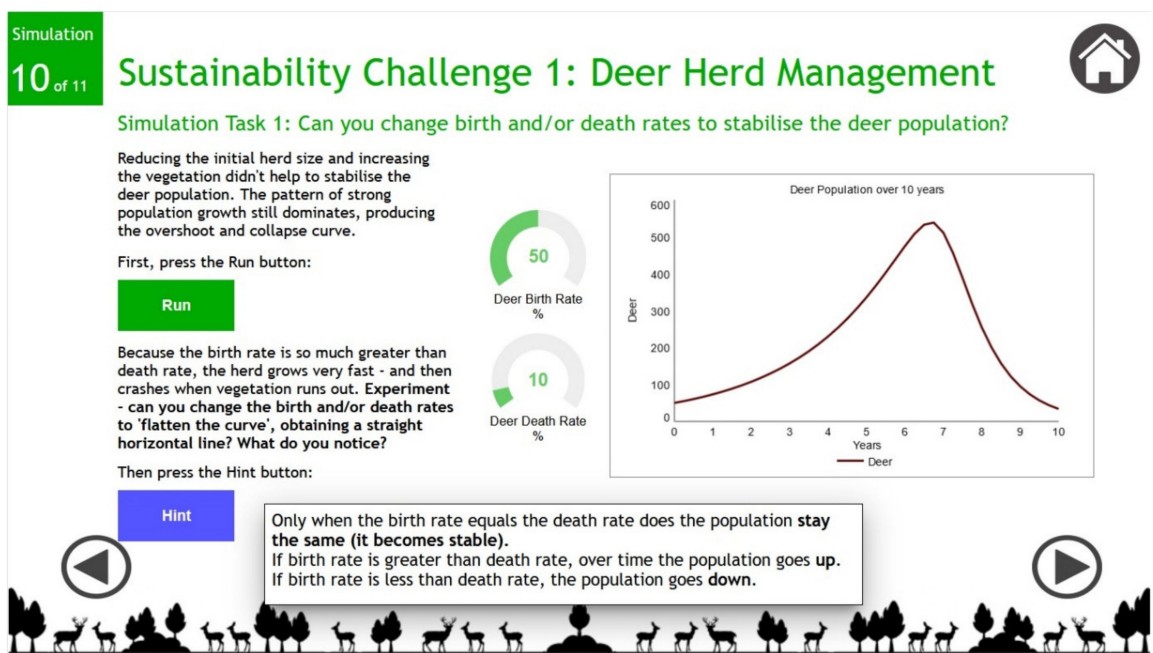

**Figure 5.** Sample screenshot of the system dynamics simulations section for exploration of deer herd management problem.

### 3.2. Case Studies, Systems Archetype and System Dynamics Model

The catastrophic unsustainable growth of the Kaibab deer herd in the US in the 1920s has been the subject of analysis by system dynamicists including Andrew Ford [31] (p. 267) and Donella Meadows (her lecture entitled 'System Dynamics Model: Kaibab Deer Population' is available online. https://www.youtube.com/watch?v=2rUXm5b-gZM, accessed on 9 November 2021). If natural predators are removed, deer will go on breeding until they overgraze and risk exceeding the carrying capacity of their environment.

The collapse of the Grand Banks cod fishery in 1992 is a famous example of disastrous unsustainable fishing practices [66]. Once one of the richest fishing grounds in the world, in 1992 the fishery collapsed completely, devastating the local community and economy. The collapse was caused by serious overfishing, which began in the late 1950s, together with poor management. Damage done to the coastal ecosystem proved irreversible and the cod fishery remains closed.

The Limits to Growth archetype, also known as Overshoot and Collapse [20] (p. 123), describes the behaviour of both these case studies well. The generic structure underlying this archetype consists of two stocks. The first stock grows exponentially while depending on a second stock, which is a renewable resource. Here, a fast-growing deer herd is eating ever more vegetation, and a growing fishing industry is exploiting fish stocks more and more heavily. This systemic structure will tend to cause the following behaviour. The first stock grows so rapidly that it overshoots, depleting the resource more rapidly than it can renew itself, leading to the collapse of the resource, and then the stock that depends on it. The deer herd overgrazes, causing collapse of the vegetation supply and then the herd. The fishing industry overfishes, so that the fish population cannot reproduce itself, destroying the industry.

The remedy for this problematic dynamic is that the exponential growth of the first stock must be checked, so that the resource on which it depends will not be consumed faster than it can regenerate. If limits (e.g., carrying capacity or maximum sustainable yield) are respected, then the system can become sustainable, meaning that the second stock, the renewable resource, remains available to the first stock indefinitely because it is not overexploited and there is time for it to renew itself. It is important to note that a description of this strategy for sustainability was seen by all participants, including the control group (although the term 'stock' was not used in the standard, non-systemic description).

The system dynamics deer herd population model used in the learning tool is slightly adapted from that documented by Breierova from the MIT System Dynamics in Education Project [65]. It is available in the Zenodo dataset published for this study.

### 3.3. Simulation-Based Learning Environments Design Principles

The field of sustainability education can benefit from the accumulated body of knowledge relating to design aspects and best practice in the development of simulation-based learning environments (SBLEs) more generally.

Landriscina advises that learners need guidance with simulations in the form of explanations, background information, tasks to perform, hints and feedback [67]. Kopainsky and Sawicka [54] (p. 143) cite Yasarcan [68], who holds that a 'gradual-increase-in-complexity approach helps improve performance in an inventory management simulation game'. In their critical review of 61 studies to evaluate the effectiveness of simulations used for science instruction, Smetana and Bell report that 'simulations used in isolation were found to be ineffective', and that they should encourage reflection and promote cognitive dissonance, meaning that learners confront their erroneous assumptions and reconstruct their beliefs [69]. Cannon-Bowers and Bowers identified the importance of using case studies as a context for instruction and setting goals for the learner [70]. Prado et al. find that both simulation and the use of case studies are effective for teaching sustainability [71].

Ghaffarzadegan et al. [72] argue that simulations based on small system dynamics models offer advantages for learning in a public policy context. By small models they mean 'models that consist of a few significant stocks and at most seven or eight major feedback loops'. These small models can 'yield accessible, insightful lessons for policy making' without overwhelming participants with too much detail.

There are two main approaches to simulation-based learning: learning by building a simulation, or by using an existing one. Reimann and Thompson assert that while learning by modelling may result in better long-term learning outcomes, positive results have also been found in studies examining the effect of learning with pre-built models [73] (p. 115). Gobert and Buckley concur [74], stating that learners can gain more insight from building models, but considerable time and skills are required. If this is not feasible, manipulation of an existing simulation offers an alternative. The approach can vary from the simplest, where learners can change a few variable values and see the consequences of their decisions on graphs, to the more complex, where learners can restructure the model. Reimann and Thompson believe that, given the greater amount of time needed to train students to use modelling software, and for them to produce a working model, 'learning with pre-built models may be a more realistic option in an environmental education context'.

The ESD learning tool developed for this study was designed in line with these general guidelines. It was designed for a single online learning session, and therefore interaction with the simulation model was limited to manipulation of a few key variables.

### 3.4. Sustainability Principles

The sustainability principles and topics listed in Table 1 and tested in the quizzes were selected from the general literature on sustainability [16,75], renewable resource management [31], and a systems view of sustainability [76] (p. 214). They were chosen as necessary skills for analysing the two cases under consideration, guided by Harris [66] for the Grand Banks fishery collapse and Meadows' analysis of the Kaibab deer dynamics in her lectures, already cited. This list forms the framework for operationalising sustainability understanding using quiz 1 (deer management) and quiz 2 (fisheries), making use of marking schemes to obtain quantitative percentage scores.

Note that these topics are limited to the cognitive aspect of sustainability understanding, not the affective, behavioural or other aspects [28].

### 3.5. Systems Thinking Principles

The essential systems thinking concepts, tools and techniques listed in Table 1 were chosen from the literature [9,77,78] as suitable tools for the analysis of the two sustainability problems under consideration.

### 3.6. Simulation Exercises

Simulations of the deer herd population model provided the basis for a series of six exercises and two tasks, listed in Table 1, which explore the dynamics that lead to overshoot and collapse, and how those dynamics can be changed so that the herd size can become sustainable. The exercises explore in stages the interplay between the two stocks, deer and vegetation, and the key role of deer birth and death rates and vegetation regeneration and consumption rates. The exercises increase in complexity as the first stock and then the second stock is added, then the interaction between the two stocks is considered, then learners are given control of key variables so that they can explore their effects on the dynamics of the deer herd.

The aim of this sequence of challenges is to demonstrate that a sustainable deer population can result if the birth and death rates are balanced and the population remains within the carrying capacity of the available land, so that the herd size will remain stable and will consume no more than the regenerated vegetation. Alternative strategies that might seem attractive, such as starting with a lower population or increasing the amount of vegetation (or effectively the size of the park), are shown to be ineffective, since the powerful exponential deer population growth dynamic dominates and will reach the limits of the park, albeit a little later. This exponential growth behaviour is seen to persist as long as the birth rate is greater than the death rate. This learning process is designed to encourage reflection, confronting erroneous assumptions and reconstructing beliefs.

### 3.7. Platform

Figure 6 shows the architecture of the learning tool. A gateway web page was used to allocate users to groups randomly and to provide a link to a Stella Architect interface with authentication and data collection enabled. The group ID passed to the Stella interface determined conditional pathways according to group. The Stella interface was published to the ISEE Exchange. Quizzes and surveys were embedded in the learning tool using SurveyMonkey surveys, and these employed custom variables to allow user identification taken from Stella logins.

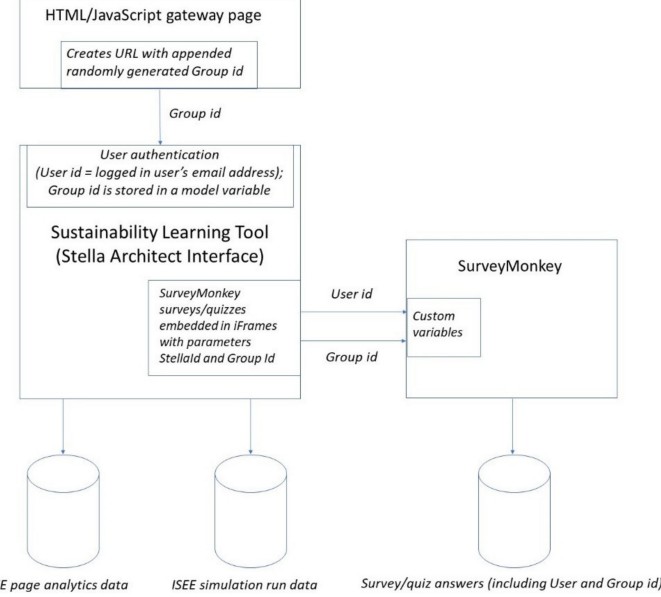

**Figure 6.** Learning tool architecture including data collected.

## 4. Methodology

### 4.1. Hypothesis and Research Questions

The general hypothesis underlying our research was:

Incorporating systems thinking increases the effectiveness of sustainability education.

The specific research questions were:

**RQ1:** Does systems thinking enhance the learner's practical understanding of sustainability?

**RQ2:** Does interacting with system dynamics simulations enhance the learner's practical understanding of sustainability?

**RQ3:** Does adding both systems thinking and system dynamics simulation enhance learning more than systems thinking only, simulation only, or a non-systemic treatment?

**RQ4:** Do systems thinking and/or simulation support the transfer of sustainability understanding from one problem to another with a similar systemic structure?

A brief account of the initial design of the study was published before the study was conducted [79]. A fuller account of the design, together with results and analysis, are all documented in the following sections. The study was conducted in the summer of 2020.

### 4.2. Study Design

The study concerned comparison of educational outcomes; therefore, the design was drawn from established practices in Social Sciences research [80]. The investigation was an experimental study using a two-by-two factorial design. The two factors, systems thinking and simulation, each had two levels: present or absent. To answer the research questions, the study aimed to discover the main effects, i.e., the effect of each factor on the learning outcome, and the interaction effect, or the combined effect of both factors.

Participants were randomly assigned to one of four groups: a control group, a systems thinking (ST) group, a simulation (Sim) group, and a systems thinking and simulation (ST + Sim) group (see Table 2).

**Table 2.** A two-by-two factorial design resulting in four experimental groups.

| Factors | No Systems Thinking | Systems Thinking |
|---|---|---|
| **No Simulation** | Control group | ST group |
| **Simulation** | Sim group | ST + Sim group |

They were then given access to the learning tool. The control group saw only standard, non-systemic content. The other groups saw additional content according to their group, either a systems thinking section, a simulation section, or both. All groups took the same two quizzes, and the performance of the groups in these quizzes was compared using statistical methods. Treatment groups were also asked to provide subjective feedback on the systems thinking and simulation features in short surveys. See Figure 7 for an overview of the research procedure.

### 4.3. Conditional Pathways for Treatment Groups

The learning tool was divided into two sections, facilitating two experiments (see Figure 8). Experiment 1 was concerned with the effect of systems thinking and/or simulation on sustainability learning outcomes, and was designed to answer RQs 1, 2 and 3. Experiment 2 was concerned with the transfer of sustainability understanding from the deer problem to the fisheries problem and was designed to answer RQ4. Quiz 1 data were captured for experiment 1, and quiz 2 data for experiment 2.

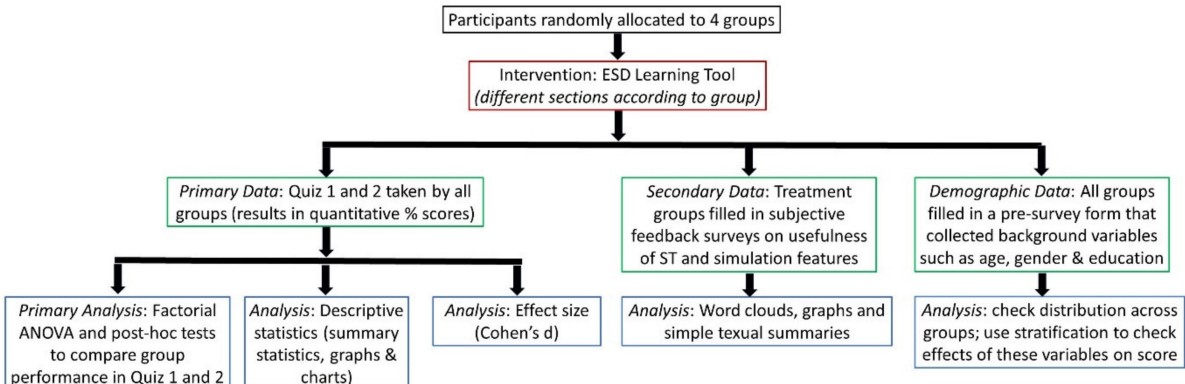

**Figure 7.** Overview of research procedure.

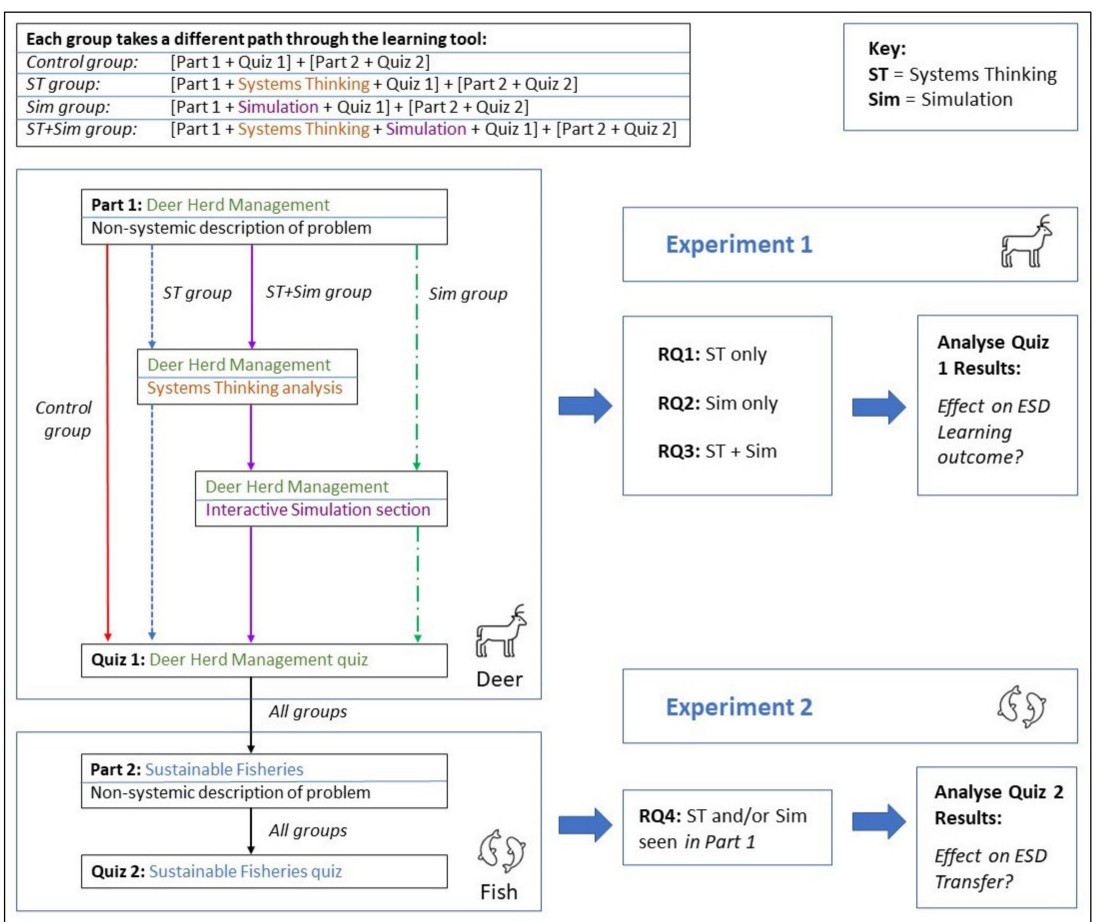

**Figure 8.** How the pathways through the learning tool and the experiments were designed to answer the research questions.

In experiment 1, a significant increase for non-control group members in quiz 1 performance would suggest that systems thinking and/or simulation improved sustainability learning outcomes (RQ1, RQ2 and RQ3).

In experiment 2, a significant increase for non-control group members in quiz 2 performance would suggest that insights from systems thinking and/or simulation applied to the deer problem resulted in a transfer of sustainability skills to the fisheries problem (RQ4), since only a standard non-systemic description of the fisheries problem was provided.

### 4.4. Teaching Method

The learning tool was originally planned to be used in a small group classroom context, with the researcher delivering an overview to the whole group before each participant would then engage with the learning tool individually. The researcher would have been available in person to answer questions about how to use or navigate the tool or to resolve any technical issues that might have arisen. However, due to COVID-19 restrictions, the training was re-designed as a single online unsupervised individual session. Support was available from the researcher via email.

### 4.5. Participants and Sampling Methods

According to UNESCO, ESD is necessary for all 'citizens, voters, workers, professionals, and leaders' [81]. This is a very large population globally, so random selection was not possible because of resource and access constraints. Subjects were instead selected using non-probability sampling techniques: a combination of two forms convenience sampling with self-selection [80] (p. 113). Convenience sampling means that participants chosen were those most easily accessible. Invitations to members of the public over the age of 18 were sent out through emails, social media or website invitations, word of mouth, etc. Individuals and groups targeted included university student societies, postgraduate students, environmental organisations and political parties, friends, acquaintances and colleagues.

Those contacted were also invited to pass the invitation on to others. This is known as snowball sampling and is a form of convenience sampling. In this way, the sample was extended, repeating until the required number of valid datasets was collected. Those who signed up were self-selected from this large network. A two-by-two factorial design requires a minimum of 20 participants per group [82] (p. 87), so at least 80 subjects were needed to be recruited.

Since the COVID-19 restrictions led to unsupervised online use, this meant that people could participate from anywhere in the world.

Randomisation was carried out by random assignment. Whilst it is a valid method for cancelling out the effects of extraneous variables, random assignment reduces generalisability across populations when compared to random selection.

### 4.6. Data Collection, Validation and Anonymisation

All quizzes and surveys were refined by pilot testing. They are openly available along with the study data in the Zenodo dataset (URL: https://zenodo.org/record/5569508 (accessed on 29 December 2021)).

The following data were collected from participants:

- In the pre-survey, basic information such as age, gender, degree subject and/or occupation, and prior knowledge of sustainability.
- Quiz 1 and quiz 2 answers comprised a mix of quantitative and qualitative data, for example, numeric answers to questions about population growth, and textual answers to questions about the meaning of sustainability in context. Each quiz question was scored numerically and included in the overall percentage results.
- The short surveys, appropriate for each treatment group, collected subjective feedback about the usefulness of the simulation and systems thinking sections. At the end of quiz 2, all participants were also asked for optional overall feedback about the learning tool.
- The email address used to log in to the learning tool was captured by ISEE and used to allow identification of survey, simulation and page analytics data.
  - Simulation data were used to verify that users had interacted with the simulation exercises.
  - Learning tool page analytics were used to judge whether participants engaged adequately with the learning tool.

Once the data were collected, datasets were validated. Validation rules used to define acceptable engagement with sections and delay in recording quiz answers are detailed in the

codebooks available in the Zenodo dataset. Datasets were also checked for completeness, according to the surveys and quizzes expected for each group. Participants were asked to promptly complete any feedback surveys that failed to record. Some cross-checking was necessary between SurveyMonkey data and ISEE data where items of data failed to record, for unknown reasons.

In this study, since the researcher knew some participants personally and had access to demographic data collected in the pre-survey, there was a risk of rater bias [83] (p. 209). Login email IDs in the data were replaced with anonymised participant IDs to avoid rater bias. The researcher may also have been influenced, consciously or unconsciously, by knowing the participant's treatment group. To reduce these risks, quizzes were marked 'blind', i.e., the researcher did not know the participant's identity nor which group they were allocated to.

Quiz scores were calculated and background variable values were recorded using predetermined marking schemes and scales. Quiz answers, marking schemes and code books for recording results are all included in the published Zenodo dataset.

*4.7. Data Analysis*

The primary analysis was inferential testing of quiz 1 and quiz 2 results. A Factorial ANOVA is an appropriate overall test for exploring the causal relationship between the two categorical independent variables and one quantitative dependent variable [84]. It detects whether any group differs significantly from the others. Factorial ANOVA differs from the standard ANOVA test, in which there is only one independent variable. If the overall test finds that there is a difference between the groups, individual post hoc tests can be conducted to find which groups differ. Certain assumptions about the distribution of the data must be met in order to conduct either form of ANOVA test. In this study, because some datasets did not fulfil the normality assumption, the Kruskal–Wallis non-parametric overall test was sometimes used instead of the standard ANOVA, followed by non-parametric post hoc tests: unpaired two-sample Wilcoxon tests. An independent two-sample t-test was also conducted to compare two groups. The rationale for selecting these tests depended on normality of the data and is outlined in the Results section.

Each significance test result in a p-value, but arguably this does not measure the strength of the relationship. An effect size such as Cohen's d is a useful complement [85]. Cohen provided basic guidelines for interpreting the effect size, namely 0.2 as small, 0.5 as medium, and 0.8 as large [86]. However, he advised that his benchmarks were recommended for use only when no better basis is available. In education research, the average effect size is d = 0.4, with 0.2, 0.4 and 0.6 considered small, medium and large effects, respectively [87].

Randomisation in the study design aims to generate comparable groups to eliminate the effect of extraneous variables, but it is always possible that unidentified confounding variables exist, confounding can be introduced by inappropriate adjustments, and the effects of confounders may not be entirely removed [88]. The approach taken to analysing extraneous variables in this study was, where known, to check their actual distribution across groups, to see if this was even, and/or to examine their effect on scores, using stratification. Data were not formally adjusted to compensate for their effects, if found; instead, sometimes adjustments were estimated, but more generally, limitations to findings and recommendations for further experimental studies were reported.

Descriptive summary statistics were also derived for quiz 1 and quiz 2 scores. For the subjective feedback on the systems thinking and simulation features, word clouds, graphs and simple textual summaries were created.

The statistical programming language R was used to create descriptive statistics such as graphs and summary statistics, to check assumptions for parametric tests, to carry out all the inferential statistics tests, to calculate effect sizes, and to analyse the effect of possibly confounding variables [89]. The R scripts necessary to reproduce all the results in detail

are openly available in the Zenodo dataset (URL: https://zenodo.org/record/5569508, accessed on 29 December 2021).

## 5. Results

The results reported in this section comprise the five types of analysis summarised in Figure 7. The section begins with a profile of the participants, summarising the background variables collected in the pre-survey. Descriptive statistics summarising quiz 1 and quiz 2 performance for all groups then follows. Analysis of possibly confounding background variables is briefly reported under the quiz 1 scores to help explore reasons for the poor performance of the full treatment group. The step-by-step procedure followed for the primary analysis, inferential statistical testing beginning with Factorial ANOVA, is then described, together with explanatory notes. Effect sizes are then reported, followed by results of the analysis of subjective feedback regarding the learning tool and its systems thinking and simulation features.

### 5.1. Profile of Participants

Of the 227 people who signed up to participate, 80 did not follow up, and 8 started but withdrew. There were 33 incomplete or invalid datasets. Some participants experienced technical issues or otherwise needed support to complete the learning experiment. After data validation, there were 106 complete datasets, one dataset per participant.

The majority of participants (58.5%) were female, and 41.5% were male. The average age was around 50 years. The great majority (85%) resided in the Ireland or the UK (60% and 25%, respectively), and 15% elsewhere. Most were graduates or postgraduates (77%), the average being a little below Master's degree level. The majority (62%) of participants had little or no prior knowledge about sustainability. The vast majority (87%) of participants had no prior knowledge of systems thinking or system dynamics. For nearly two-thirds (64%) of participants, their occupation and/or education had no relevance or little relevance to sustainability or systems thinking.

### 5.2. Descriptive Statistics

5.2.1. Experiment 1: Quiz 1 (Deer Herd Management) Scores

Results are shown in Table 3. The simulation group performed best, with the highest mean and median scores. All treatment groups performed better than the control group. A boxplot showing the distribution of quiz scores is shown in Figure 9. This shows an outlier in the control group.

**Table 3.** Deer Herd Management Quiz (Quiz 1) Scores by Group.

| Group | Control | ST | Sim | ST + Sim | All |
|---|---|---|---|---|---|
| Total participants | 28 | 26 | 24 | 28 | 106 |
| Min% score | 40 | 55 | 48 | 46 | 40 |
| Max% score | 92 | 97 | 96 | 97 | 97 |
| Mean% score | 70.8 | 75.8 | 78.4 | 72.9 | 74.3 |
| Median% score | 73.5 | 76.5 | 84 | 77 | 77 |
| Standard deviation | 11.0 | 11.3 | 14.1 | 11.8 | 12.2 |
| Outlier score(s) | 40 | - | - | - | - |

The mean score of the full treatment group (ST + simulation) was lower than that of the systems thinking group and the simulation group, which was unexpected. The question arose, why was the mean score obtained when both factors were combined not at least as high as that obtained with either of the factors alone?

Scores for individual quiz questions were compared to find out which groups performed best on specific sustainability topics. The ST group outperformed other groups in questions about maximum capacity. The Sim group outperformed other groups in

questions about the definition of sustainability, identifying sustainable graph patterns, calculating multiple interacting stock levels, identifying the point where limits were reached, and choosing policies for sustainability.

**Figure 9.** Boxplot of Quiz 1 Scores by Group.

5.2.2. Possible Reasons for Poor ST + Simulation Group Performance

The ST + Sim group obtained lower than expected results. The distribution of known background variables between groups was explored in case these were confounding, and this group was found to differ from the others in three ways. There were far more participants aged over 65, they had far less prior sustainability knowledge, and there were far more delays in both quizzes due to technical issues or interruptions. However, after closer analysis using stratification, higher age and more delays were found not to be associated with lower quiz scores. A lower average prior sustainability knowledge score did affect score a little: estimating the effect on group mean score of increasing the average level of prior knowledge to that of other groups suggests an increase of 1.2%, not enough to create a significant result for that group. However, further studies could use techniques such as restriction or matching in the study design to eliminate any possible effect. A design that simply excluded people with high prior sustainability knowledge could be sufficient.

A much more likely explanation for the poor performance of this group, the full treatment group, was found in the significant negative interaction effects uncovered by Factorial ANOVA testing and described in the Inferential Statistics section.

5.2.3. Experiment 2: Quiz 2 (Sustainable Fisheries Management) Scores

Results are shown in Table 4. Again, the simulation group performed best, with the highest mean and median scores. Other treatment groups performed worse than the control group, when comparing the means, medians and modes of group scores. A boxplot showing the distribution of quiz scores is shown in Figure 10. There are two outliers in

the control group, one of which (the lowest score) belonged to the same participant as the outlier in Figure 9. A second outlier was less extreme, with a score of 59%, and was only an outlier for the control group, not the participants as a whole. No obvious errors or unusual circumstances gave rise to this outlier.

**Table 4.** Sustainable Fisheries Quiz (Quiz 2) Scores by Group.

| Group | Control | ST | Simulation | ST + Simulation | All |
|---|---|---|---|---|---|
| Total participants | 28 | 26 | 24 | 28 | 106 |
| Min% score | 38 | 55 | 55 | 54 | 38 |
| Max% score | 92 | 97 | 95 | 89 | 97 |
| Mean% score | 78.3 | 77.5 | 82.0 | 73.7 | 77.7 |
| Median% score | 80.5 | 78 | 83.5 | 73.5 | 79.5 |
| Standard deviation | 11.1 | 11.2 | 9.9 | 8.9 | 10.6 |
| Outlier score(s) | 38, 59 | - | - | - | 38 |

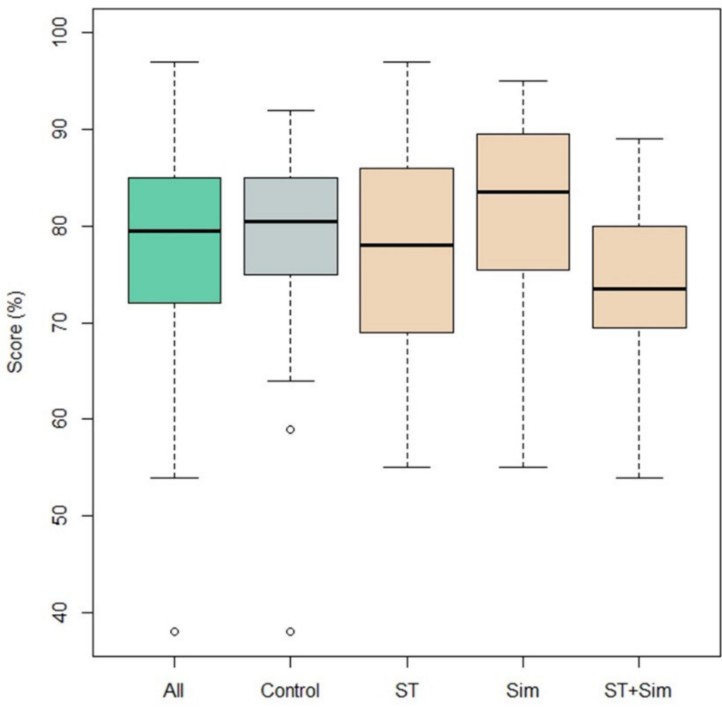

**Figure 10.** Boxplot of Quiz 2 Scores by Group.

Scores for individual questions were compared by group to find out which groups performed best on specific sustainability topics. The Sim group outperformed all other groups in the calculation of years to maximum fishery capacity, in understanding maximum sustainable yield, and in identifying sustainable graph patterns. It performed a little better than other groups in single stock exponential growth and maximum capacity calculations, and in defining sustainability in the context of fisheries.

### 5.3. Inferential Statistics

Table 5 summarises the process followed when conducting the inferential statistical tests on quiz 1 and quiz 2 data. The main findings are shown in the 'Interpretation' column. All R scripts necessary to reproduce the results outlined here, including the analysis of possible confounding variables, are available in the published Zenodo dataset. The following paragraphs provide explanatory notes.

**Table 5.** Overview of the inferential testing process, beginning with Factorial ANOVA.

| Quiz Data | Steps | Purpose | Type | Test | Result | Interpretation | Decision |
|---|---|---|---|---|---|---|---|
| Quiz 1 | 1 | Overall test on all four groups | Parametric | Factorial ANOVA | A significant result only for the interaction of ST and Sim ($p$-value 0.045), not for either of the factors alone (the main effects). | An interaction plot shows an 'antagonistic' interaction effect, very strong because the lines are nearly perpendicular. Main effects are therefore uninterpretable [82]. | Remove data for ST + Sim group. Use a one-way ANOVA (group as independent variable) |
| | 2 | Overall test on three groups (control, ST and Sim) | Non-parametric (assumptions not met for ANOVA) | Krugal–Wallis test | $p$-value 0.067, significant at the 90% confidence level. | It is likely that at least one group differs from the others. | Proceed with post hoc tests to compare ST & control, and Sim & control (2 comparisons) |
| | 3 | Post hoc tests on the above three groups | Non-parametric (assumptions not met in step 2) | Pairwise Wilcoxon rank sum tests (one-tailed) | One significant result, an unadjusted $p$-value 0.009 for the Sim group. The adjusted $p$-value of 0.018 (Bonferroni adjustment for two comparisons) is significant at the 95% confidence level. | The Sim group scored significantly better than the control in quiz 1. | - End testing - |
| Quiz 2 | 1 | Overall test on all four groups | Parametric | Factorial ANOVA | A significant result for the interaction of ST and Sim ($p$-value 0.052) at the 90% confidence level. | An interaction plot shows an 'antagonistic' interaction effect. The lines cross and both slope downwards. The downward slope shows that ST always brings down the score. | Only Sim group performed better than control. Proceed to compare these. |
| | 2 | Compare Sim and control groups | Parametric | Two-sample independent t-test (one-tailed) | $p$-value 0.079, significant at the 90% confidence level. | Sim scores were better than the control group scores in quiz 2, but the result is weaker than for quiz 1. | - End testing - |

From the literature, there is a clear expectation that systems thinking and/or simulation will increase understanding of sustainability problems. Therefore, alternative hypotheses tested asserted that scores for treatment groups would be greater than those of the control group, leading to right-tailed (one-tailed) significance tests. The null hypotheses were that there were no differences between the groups.

Where a parametric test was conducted, the appropriate assumptions for the test were first checked. The assumptions for ANOVA tests are the independence of observations, the homogeneity of variances and the normality of residuals [89] (p. 517). The first condition is satisfied since participants in this study were randomly allocated to treatment groups. Levene's test for homogeneity of variance and the Shapiro–Wilk test for normality of residuals were both carried out using R on the appropriate datasets to check the other two assumptions. The assumptions for the two-sample independent t-test are similar: independence of the observations, an approximately normal distribution for each group, and homogeneity of variances [ibid] (p. 397). If the assumptions were not met, a suitable non-parametric test was used instead.

Parametric tests do not work well when there are outliers [80] (p. 592). The outlier score in quiz 1 was removed, and the more extreme outlier in quiz 2 was also removed before ANOVA testing. Finally, two quiz 2 datasets were removed prior to analysis, as page analytics logs revealed that these participants did not engage with the fisheries section of the learning tool. They both spent no more than two minutes on the fisheries section, whereas the minimum acceptable time was 5.5 minutes, and recommended time was 15 minutes.

The Factorial ANOVA tests revealed that both in quiz 1 and quiz 2 the presence of both factors (systems thinking and simulation) created a negative, or 'antagonistic' interaction effect. Interaction plots are shown in Figures 11 and 12. This means that adding a second treatment reduced the quiz scores. The interaction effect partly cancelled out the main effects of each factor alone. This refutes RQ3.

### 5.4. Effect Size

The best performing group was the Sim group with Cohen's d effect size calculated at 0.6. This is a large effect in an educational context. ST improved learning outcomes but had a weaker effect (Cohen's d 0.4, a medium effect). ST + Sim had a still weaker effect (0.1, a very small effect).

For quiz 2, the Sim group was the only group that performed better than the control group, so the effect size for other groups was not calculated. Cohen's d was calculated at 0.4. This is a medium effect in an educational context.

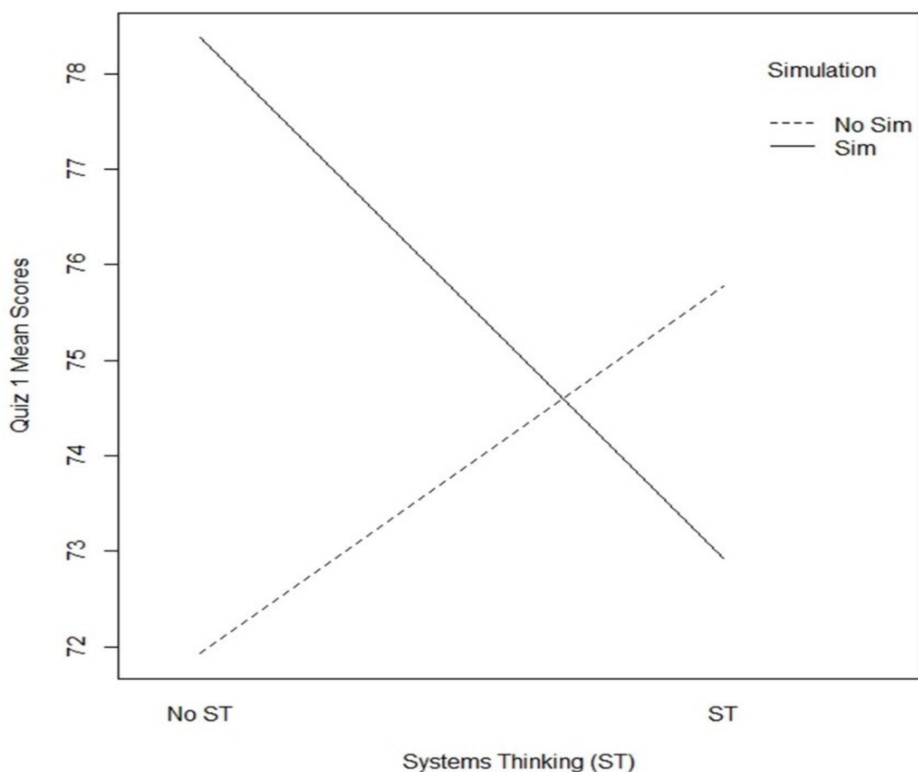

**Figure 11.** Interaction plot showing effect of both factors on quiz 1 score.

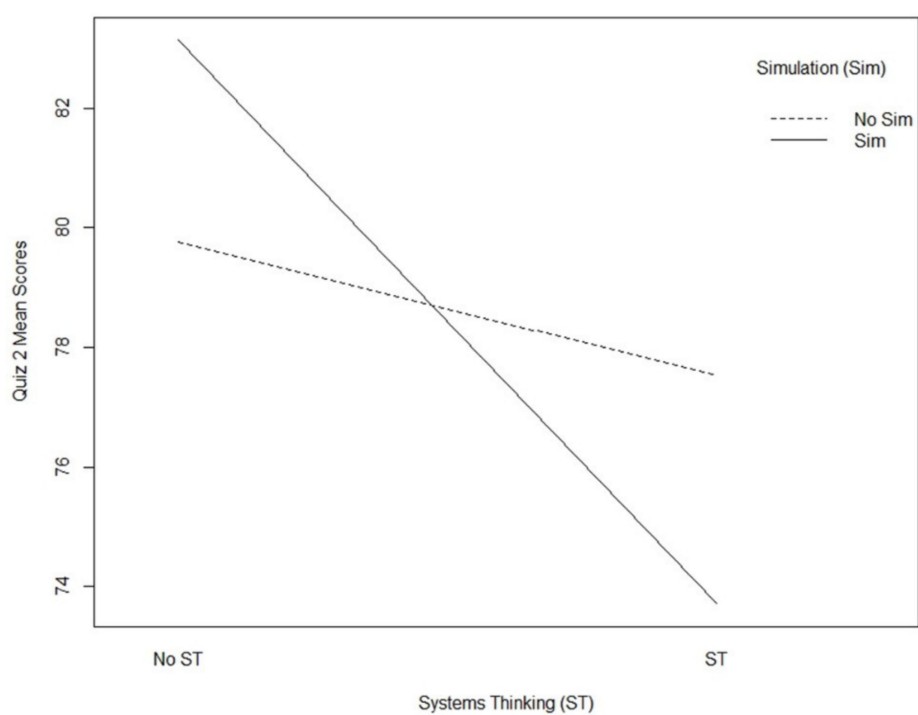

**Figure 12.** Interaction plot showing effect of both factors on quiz 2 score.

Table 6 provides a formal summary of the results of the inferential tests and effect sizes and provides answers to the research questions and main hypothesis.

**Table 6.** Summary of results of inferential testing results, with effect sizes.

| Research Question | Summary of Results |
|---|---|
| **RQ1:** | **Does systems thinking enhance the learner's practical understanding of sustainability?** |
| *Result:* *No* | A Wilcoxon rank sum test (one-tailed, and using the Bonferroni correction to adjust p) showed that there was no significant increase in mean scores for the 26 participants in the systems thinking group (M = 75.8, SD = 11.3) compared to the 27 participants in the control group ($p$ = 0.247), despite participants attaining higher scores than the control group (M = 71.9, SD = 9.3). The effect size was medium in the educational context (Cohen's d 0.4). |
| **RQ2:** | **Does interacting with system dynamics simulations enhance the learner's practical understanding of sustainability?** |
| *Result:* *Yes* | A Wilcoxon rank sum test (one-tailed, and using the Bonferroni correction to adjust p) showed that the 24 participants in the simulation group (M = 78.4, SD = 14.1) compared to the 27 participants in the control group (M = 71.9, SD = 9.3) demonstrated significantly better mean scores, ($p$ = 0.018) at the 95% confidence level ($\alpha$ = 0.05). The effect size was large in the educational context (Cohen's d 0.6). |
| **RQ3:** | **Does adding both systems thinking and system dynamics simulation enhance learning more than systems thinking only, simulation only, or a non-systemic treatment?** |
| *Result:* *No* | The 28 participants in the systems thinking and simulation group performed slightly better (M = 72.9, SD = 11.8) than the control group (M = 71.9, SD = 9.3), but worse than the other treatment groups. The effect size was very small in the educational context (Cohen's d 0.1). A Factorial ANOVA test found a significant interaction effect between the two factors on score ($p$ = 0.045). The interaction effect was negative, since the presence of one factor reduced the effect of the other. The interaction effect was therefore antagonistic to the main effects of the factors. |
| **RQ4:** | **Do systems thinking and/or simulation support the transfer of sustainability understanding from one problem to another with a similar systemic structure?** |
| *Result:* *Yes, only simulation* | The 23 participants in the simulation group (M = 83.1, SD = 8.3) compared to the 26 participants in the control group (M = 79.8, SD = 8.1) did not demonstrate significantly better mean scores at the 95% confidence level ($\alpha$ = 0.05): t(47) = 1.44, $p$ = 0.0787. However, the mean scores were significantly better at the 90% confidence level ($\alpha$ = 0.10). The effect size was medium in the educational context (Cohen's d 0.4). Other treatment groups did not perform better than the control group. A Factorial ANOVA test found a significant interaction effect at the 90% confidence level between the two factors on score ($p$ = 0.052). The interaction effect was negative, since the presence of each factor reduced the effect of the other factor. Systems thinking reduced scores. |
| **Hypothesis:** | Incorporating systems thinking increases the effectiveness of sustainability education. |
| *Result:* *Only simulation* | **Only simulation was found to significantly increase sustainability quiz scores. Systems thinking increased scores in quiz 1, but not significantly.** |

*5.5. Feedback from Participants*

The feedback summary in this section is available in greater detail [90] and the full data are published in the Zenodo dataset. A short summary follows below.

5.5.1. General Feedback on the Learning Tool

Comments were generally very positive. The most frequent words used are visualised in the word cloud in Figure 13. The most frequent evaluative words used were 'interesting' and 'informative'.

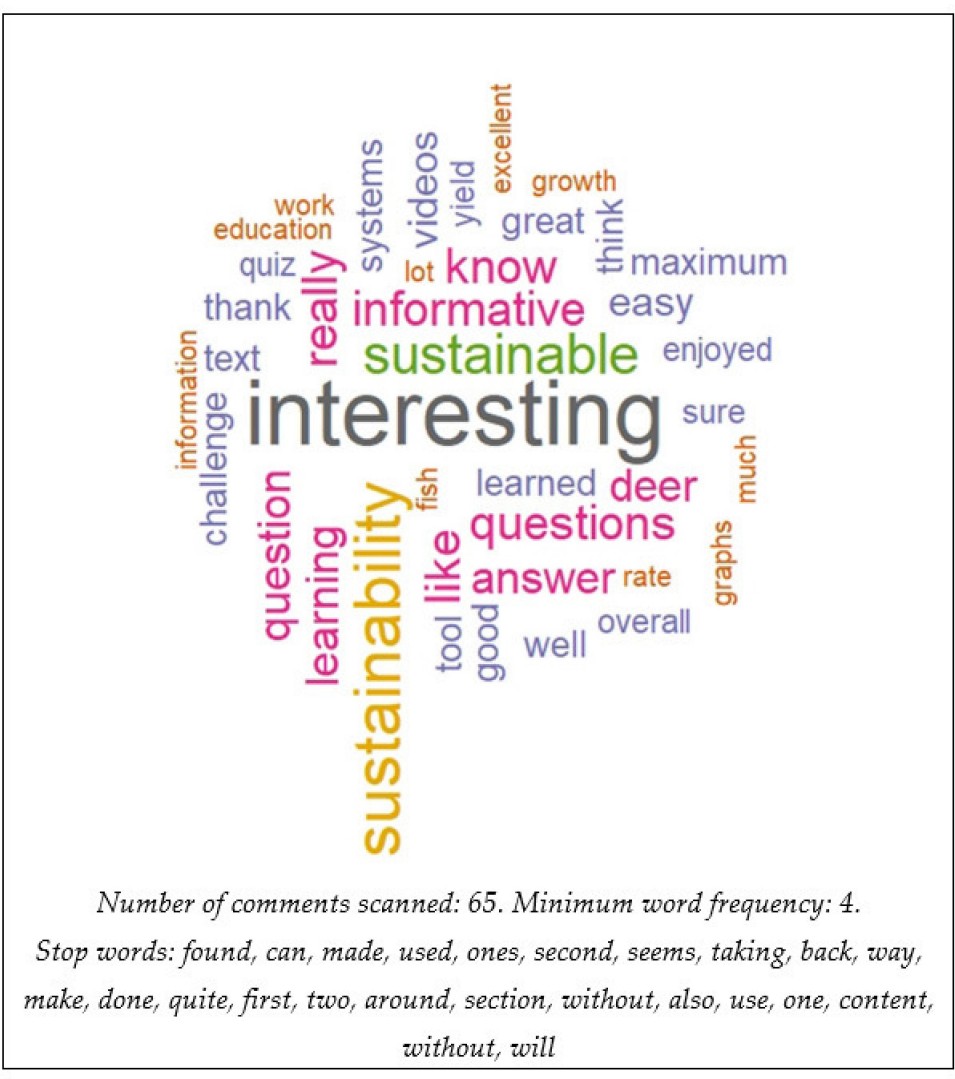

**Figure 13.** Word cloud of optional general comments about the learning tool.

Some people commented favourably on the benefits of interactive learning with simulation. It helped them better understand cause and effect and consequences of policy decisions, allowed experimentation and knowledge construction, and made learning enjoyable. Case studies were found useful. Some people found the mathematical aspects of the learning material challenging.

### 5.5.2. Feedback on the Usefulness of Systems Thinking

Participants with access to the systems thinking section were asked to rate how useful they found it on a 5-point Likert scale (see Figure 14). About three-quarters (74.1%) of participants said it helped quite a lot or really transformed the way they saw the problem.

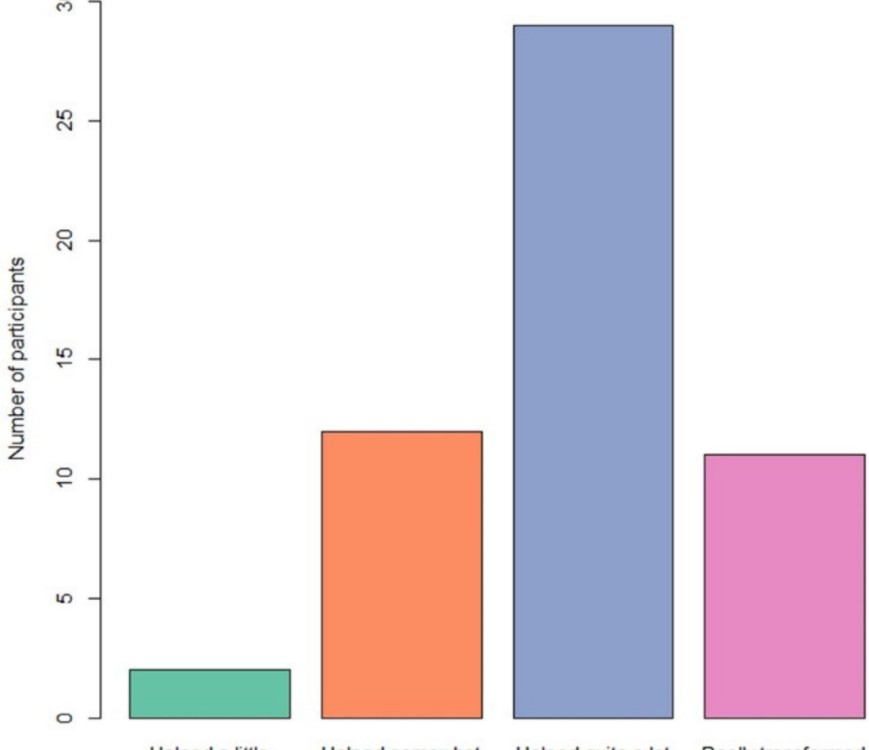

**Figure 14.** Bar graph showing usefulness ratings given for systems thinking.

Participants commented that systems thinking was useful for understanding interrelationships and how systems interlink, identifying the point at which systems become unsustainable, clarifying cause and effect, identifying patterns of behaviour and changes over time, and making decisions. A few expressed concern about remembering the complex terminology.

### 5.5.3. Feedback on the Usefulness of Simulation

Participants with access to the simulation section were asked to rate how useful they found it on a 5-point Likert scale (see Figure 15). The great majority (84.6%) of participants felt that simulation helped quite a lot or really transformed the way they saw the problem, higher even than the 74.1% of participants who felt the same about systems thinking.

Participants found simulation useful for increasing clarity and understanding by adjusting variables, experimenting with strategies, assessing impacts and informing policy decisions, for seeing how quickly resources can be depleted, for finding sustainable limits, and for teaching responsibility. Interactivity helps learning and retention, some said, and seeing graphs change dynamically is more effective than reading text for understanding complexity and real-world problems and performing the mathematical work themselves.

**Figure 15.** Bar graph showing usefulness ratings given for simulation.

## 6. Discussion and Conclusions

The main findings were that system dynamics simulation has a strong effect on understanding a sustainability problem, and a weaker but still significant effect (at the 90% confidence level) on the transfer of understanding to another problem with a similar systemic structure. Systems thinking did not make a significant difference to mean scores in either case, and the combination of systems thinking and simulation in the full treatment group had a negative effect. This could be evidence that the additional learning material, or perhaps its abstract complexity, pushed participants over a limit with respect to 'cognitive load' [91] in this experimental setting (a single learning session). It could also be evidence that quantitative simulation has a better learning outcome than more qualitative approaches. Interactive simulation provides an opportunity for learners to perform actions (operations) and build their understanding of a system through 'operational thinking' [57,92].

Feedback from participants was very positive, with a large majority reporting finding systems thinking and simulation useful. However, only for simulation was this backed up by a statistically significant increase in quiz performance.

We conclude that simulation is a powerful and highly efficient way of teaching sustainability. Results were achieved with a short (20 min) simulation section comprising a few guided tasks and exercises. Systems thinking theory in the abstract was not as powerful in its learning effect as action and interaction with simulation, and guided simulation worked *even without* explicit systems thinking theory. The systems thinking principles were communicated implicitly in the guidance and progression of themes explored in the simulation exercises.

These findings are consistent with Forrester [58] and Sterman [20], who consider simulation essential for learning systems thinking. It adds weight to various initiatives and reported benefits of simulation for environmental learning [34,47–54,93]. This has important policy implications for ESD.

Simulation is an efficient way of teaching sustainability, not only for the problem represented, but very likely for other problems with a similar systemic structure. Findings regarding the transfer of knowledge are consistent with Kumar and Dutt [94] who find that simulation helps transfer stock-and-flow learning from one problem to another. This transfer effect has the potential to make sustainability education more efficient and more pattern-based. According to Bloom, 'pattern thinking is at the core of all human thinking,

in which the brain functions as a pattern recogniser' [95]. This approach has the potential to build problem-solving ability and systems and environmental literacy.

In conclusion, simulation is a highly effective tool for enhancing sustainability understanding in a single short learning session, even when learning is performed remotely online without supervision. This finding is the major contribution of this study. Other contributions are the innovative learning tool (which is openly available, together with quizzes, answers and marking schemes), the experimental assessment framework, the formal trial centred on the learning tool, and the openly available R code used for data analysis.

### 6.1. Limitations of the Study

Our conclusions are limited to the effect of the factors in a learning environment designed for a single individual learning session. The participants, since they were not randomly selected, may not represent the whole population, suggesting that the study should be repeated to check external validity. The findings of the study are limited to cognitive aspects of sustainability understanding, not its affective or behavioural aspects.

### 6.2. Suggestions for Future Work

The medium effect size and positive feedback from learners suggests that systems thinking may be useful if presented differently. It may have a stronger effect on the understanding and development of transferrable skills if taught in an interactive classroom or group situation and not limited to a single session. Care should be taken not to cause excessive cognitive load on learners.

Furthermore, effect sizes in educational research are often categorised as small by Cohen's standards [96]. This is because there are usually many other important factors affecting results, typically prior education and skills such as numeracy, literacy, science and so on. Interpreting effect sizes in educational interventions is a complex matter and is evolving [ibid]. Where effect sizes are modest, two strategies can increase the power of the study. Firstly, the sample size can be increased. The effect size can also be increased by making the quizzes harder, since the baseline scores were rather high, making differences due to the factors harder to observe. Increasing statistical power decreases the probability of a Type II error, in which the researcher wrongly concludes that there is no effect when one actually exists [85].

Since systems thinking and simulation were the factors under investigation, and sustainability understanding was the dependent variable, an improvement to a future study design would be to exclude people already knowledgeable in those areas.

The methodology (factorial study design coupled with Factorial ANOVA significance testing) could be usefully employed for further studies to investigate the effectiveness of various styles of learning intervention as factors, such as multiple learning sessions, role play and group model building, or it could be used to evaluate existing virtual worlds, games and simulators.

The experimental framework for assessing sustainability understanding proved useful for creating a quantitative measure of sustainability understanding in the context of specific sustainability problems. It could be further refined and adapted for use with different case studies.

A simulation-based learning tool could be used to improve public understanding in other complex fields such as the dynamics of infectious disease and mitigation strategies. This could create a useful antidote, or 'psychological inoculation' [97], to the spread of misinformation that proved so harmful during the COVID-19 pandemic.

### Research Ethics Statements

All subjects were adults aged over 18 and gave their informed consent for inclusion before they participated in the study. For this non-interventional study, all participants were fully informed that confidentiality was assured, that data would be anonymised before publishing, why the research was being conducted, how their data would be used, any risks associated, and their right to withdraw.

**Author Contributions:** Conceptualization, C.G.; Methodology, C.G.; Software, C.G.; Validation, C.G.; Formal Analysis, C.G.; Investigation, C.G.; Resources, J.D. and O.M.; Data Curation, C.G.; Writing—Original Draft Preparation, C.G.; Writing—Review and Editing, O.M. and J.D.; Visualization, C.G.; Supervision, J.D.; Project Administration, J.D.; Funding Acquisition, J.D. and O.M. All authors have read and agreed to the published version of the manuscript.

**Funding:** This research was undertaken for the PhD studies of the corresponding author at the National University of Ireland Galway (NUIG) and was supported by funding from ResponSEAble (EU Horizon 2020 project number 652643), Ireland's Higher Education Authority and Department of Further and Higher Education, Research, Innovation and Science (through the IT Investment Fund and ComputerDISC, and the COVID-19 Costed Extension), and the NUIG PhD Write-Up Bursary.

**Institutional Review Board Statement:** Not applicable.

**Informed Consent Statement:** Informed consent was obtained from all subjects involved in the study.

**Data Availability Statement:** Data are available in a publicly accessible repository. The anonymised data presented in this study, together with the R Scripts necessary to reproduce all results, are openly available in Zenodo. DOI: https://doi.org/10.5281/zenodo.5569508; URL: https://zenodo.org/record/5569508.

**Conflicts of Interest:** The authors declare no conflict of interest.

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
