# Peer review of "An Empirical Study of the Impact of Systems Thinking and Simulation on Sustainability Education"

_sustainability, doi:10.3390/su14010394_

Round 1
Reviewer 1 Report
The paper highlights the role of improving pedagogical methods to tackle sustainability challenges. The study investigates the effect of system thinking on education for sustainable development. A control trial study is conducted to examine the influence of online sustainable learning tools on increasing the learning outcomes of a specific problem. The online tool incorporates systems thinking and system dynamics simulation. The paper aim is clear and justified within the literature. The hypothesizes are well defined and the study method is properly tailored to investigate the hypothesis. The dear herd and fisheries management cases are ideal choices for investigating the hypotheses where they are broadly studied in the literature. The result is sufficiently analyzed and discussed, and valuable conclusions are deduced.
However, I have some suggestions to improve the paper. Although the paper is written in good language, the paper structures could be improved to be in line with matured and widely followed structure in the scientific community as well as for better readability. The first impression you get when you read the abstract and the introduction is that the manuscript is not written for a journal article. The abstract is very long and should be structured in only one paragraph. The abstract should briefly state the purpose of the research, the method, the result, and major conclusions. These should be stated in only one paragraph, not three.
The introduction is very long and does not convey the whole story of the paper. Including the review of the related work in the introduction does not help the paper readability. There are lots of articles that merge the literature to the introduction, but this is because of the limited number of pages, and it is condensed and well structured, delivering the introduction message. However, the MDPI journals have the privilege of no page’s limits. Also, the introduction is very long and missing the main objective which is to provide a short and complete story of the article. The introduction usually states the context of the work, the gap, the contribution, the objective of the work without a detailed literature survey. A detailed literature survey should be conducted in a separate section to detail the mentioned gap in the introduction.
The link to the learning tool at https://exchange.iseesystems.com/public/carolineb/sustainability-learning-tool is not working.
Author Response
Thank you for your careful review of our manuscript. We respond point-by-point below.
Point 1: The abstract is very long and should be structured in only one paragraph. The abstract should briefly state the purpose of the research, the method, the result, and major conclusions. These should be stated in only one paragraph, not three.
Response 1: I have reworked the Abstract according to your instructions. The word count falls within the 200 word limit given in the author guidelines.
Point 2: The introduction is very long and does not convey the whole story of the paper. Including the review of the related work in the introduction does not help the paper readability. There are lots of articles that merge the literature to the introduction, but this is because of the limited number of pages, and it is condensed and well structured, delivering the introduction message. However, the MDPI journals have the privilege of no page’s limits. Also, the introduction is very long and missing the main objective which is to provide a short and complete story of the article. The introduction usually states the context of the work, the gap, the contribution, the objective of the work without a detailed literature survey. A detailed literature survey should be conducted in a separate section to detail the mentioned gap in the introduction.
Response 2: I followed the author guidelines closely (https://www.mdpi.com/journal/sustainability/instructions#manuscript). The advice seemed to be that only Introduction, Materials and Methods, Results, Discussion, and optionally, Conclusion sections are required for research manuscripts.
I think your suggestion of having a separate Literature Review section really improves the clarity of the paper and I have done this. I have also improved the Introduction to summarise the article, as you have suggested. I also moved the material about the learning tool to its own section, after the Literature Review, which, I hope, structures the article better and makes it easier to read and understand.
Point 3: The link to the learning tool at https://exchange.iseesystems.com/public/carolineb/sustainability-learning-tool is not working.
Response 3: Thank you for pointing this out. The specific link is now fixed. I also checked all links in the paper to make sure they are working live links.
Reviewer 2 Report
Dear authors,
you made an effort to present the results of your research. However, the topic is not presented in a comprehensive way. You have too many pages, too many details and the meaning is lost for the reader in some pages. The reader also loses his interest and finally does not understand the originality of your paper. In addition, your literature is that old. You have 66 references and not even half of them are of the three last years.
Author Response
Thank you for your careful review of our manuscript. We respond point-by-point below.
Point 1: You have too many pages, too many details and the meaning is lost for the reader in some pages. The reader also loses his interest.
Response 1: I have removed some of the detail (e.g. some of the tool screenshots, details about surveys and quizzes, and figures and details in the ‘Feedback from Participants’ section).
It was difficult to balance the request to cut pages with requests from other reviewers, who recommended breaking the paper up into more sections adding section summaries to guide the reader, and expanding the discussion section. This has resulted in some additional text, but I believe the better structuring and guidance will mean that the reader will navigate the details.
I have striven to find the right balance between the conflicting requests of the different reviewers, so that now the restructured paper should hold the reader’s interest better.
Point 2: The reader finally does not understand the originality of your paper.
Response 2: The research gaps and contributions are now more clearly summarised in the more compact introduction, explicitly stated at the end of the Literature Review section, and revisited with conclusions in the ‘Discussion and Conclusions’ section.
Point 3: Your literature is old. You have 66 references and not even half of them are of the three last years.
Response 3: I have added 26 new references, 21 from the last 3 years. I think this has improved the paper.
However, older references remain. The purpose of the paper is to provide a review of decades of work in the System Dynamics and Systems Thinking fields to inform the much newer Sustainability Education field, and I have chosen many of the most important authors and seminal works in these longer-established fields which stand to this day.
Our paper is intended to be a comprehensive review which we genuinely hope will be of service to those in the ESD field who are currently urgently calling for effective systems-based educational tools and techniques.
There are also seminal works included from the field of statistics, which are important for justifying the methodology used.
Reviewer 3 Report
It is a great paper with a significant contribution to the current literature, I strongly recommend publishing this paper.
Author Response
Dear reviewer,
Thank you for taking the time to review our paper, and for your positive and encouraging feedback.
Kind regards,
Caroline Green
Reviewer 4 Report
First of all I would like to pay attention to the incorrect format of the article.
- Abstract has inappropriate structure. I suggest to answer the following aspects: - general context - novelty of the work - methodology used (describe briefly the main methods or treatments applied) - main results and related interpretations.
- Introduction: This section should briefly place the study in a wide context and emphasize why it is relevant carrying out the analysis. It should define the purpose of the work and its significance. In this perspective, this section is too succinct and fails to effectively point out the relevance of your contribution towards the existing literature. Moreover, the authors do not provide at the end of the section the description of the paper structure which is very useful for readers.
- Literature Review: This chapter is important. The authors present a rather modest system of analysis that can be further improved. It would be useful to analyze more and new sources.
- The research methodology seems underdeveloped. Methods should be described in detail. I think the research procedure could be much more clearly described by means of a diagram also highlighting its potential and limit. The article is full of tables and figures, but I lack a more detailed explanation.
- Results are not always linked to the methodology. Please define the relationship and relate your finding with the relevant literature.
- The discussion part is succinct. Authors should disclose their essential “discoveries”. I would suggest the authors to frame it as a "typical" conclusive section. Please provide limitation, future research needs as well as practical / policy implications.
Author Response
Thank you for your very careful review of our manuscript. We respond point-by-point below.
Point 1: Abstract has inappropriate structure. I suggest to answer the following aspects: - general context - novelty of the work - methodology used (describe briefly the main methods or treatments applied) - main results and related interpretations.
Response 1: I have reworked the Abstract according to your recommendation. The word count falls within the 200 word limit given in the author guidelines.
Point 2: Introduction: This section should briefly place the study in a wide context and emphasize why it is relevant carrying out the analysis. It should define the purpose of the work and its significance. In this perspective, this section is too succinct and fails to effectively point out the relevance of your contribution towards the existing literature. Moreover, the authors do not provide at the end of the section the description of the paper structure which is very useful for readers.
Response 2: I have created a new, concise Introduction according to your recommendations. Most of the material that was previously in the introduction is now in the Literature Review section.
Point 3: Literature Review: This chapter is important. The authors present a rather modest system of analysis that can be further improved. It would be useful to analyze more and new sources.
Response 3: I followed the author guidelines closely (https://www.mdpi.com/journal/sustainability/instructions#manuscript). The advice seemed to be that only Introduction, Materials and Methods, Results, Discussion, and optionally, Conclusion sections are required for research manuscripts.
I think your suggestion of having a separate Literature Review section really improves the clarity of the paper and I have done this. I have also added 26 new sources to the paper, mostly in this section.
Point 4: The research methodology seems underdeveloped. Methods should be described in detail. I think the research procedure could be much more clearly described by means of a diagram also highlighting its potential and limit. The article is full of tables and figures, but I lack a more detailed explanation.
Response 4: I have added a new diagram to describe the research procedure (Figure 7, Overview of research procedure). This shows more clearly the three main types of data collected and the five main types of data analysis conducted. I have also rewritten parts of the Data Analysis section in the Methodology section to clarify the methodology. I have also moved the sections on the ESD learning tool out of the methodology section into a separate section, for greater clarity.
Point 5: Results are not always linked to the methodology. Please define the relationship and relate your finding with the relevant literature.
Response 5: I have restructured the first part of the ‘Results’ section by relating the various results reported to the five main types of data analysis shown in the new diagram (Figure 7). I have also worked on improving the clarity of the ‘Inferential Statistics’ section.
I have endeavoured to explain that the process of choosing appropriate inferential tests began with Factorial ANOVA testing, as planned, but that, as shown in Table 5, the characteristics of the data affected the choice of tests. For example, as reported in the first row of Table 5, the negative interaction effect revealed by Factorial ANOVA necessitated removing the data for the full treatment group, but remaining data (see row 2) was not suitable for ANOVA testing as it did not fulfil the normality assumption, so a non-parametric alternative test (Krugal-Wallis) was selected.
I have added new text to the ‘Discussion and Conclusion’ section to relate our findings to the relevant literature.
Point 6: The discussion part is succinct. Authors should disclose their essential “discoveries”. I would suggest the authors to frame it as a "typical" conclusive section. Please provide limitation, future research needs as well as practical / policy implications.
Response 6: We have clarified the ‘Discussion and Conclusion’ in line with your recommendations, and we feel it is now much improved.
Round 2
Reviewer 2 Report
It is obvious that you have considered the comments and made an improvement to the manuscript.
Author Response
Thank you for your comments, which really helped improve the paper.
Reviewer 4 Report
Accept in present form.
Author Response
Thank you for your thorough comments, which really helped improve the paper.